# Circulating Low Density Neutrophils Are Associated with Resistance to First Line Anti-PD1/PDL1 Immunotherapy in Non-Small Cell Lung Cancer

**DOI:** 10.3390/cancers14163846

**Published:** 2022-08-09

**Authors:** Hugo Arasanz, Ana Isabel Bocanegra, Idoia Morilla, Joaquín Fernández-Irigoyen, Maite Martínez-Aguillo, Lucía Teijeira, Maider Garnica, Ester Blanco, Luisa Chocarro, Karina Ausin, Miren Zuazo, Gonzalo Fernández-Hinojal, Miriam Echaide, Leticia Fernández-Rubio, Sergio Piñeiro-Hermida, Pablo Ramos, Laura Mezquita, David Escors, Ruth Vera, Grazyna Kochan

**Affiliations:** 1Oncoimmunology Group, Navarrabiomed, Instituto de Investigación Sanitaria de Navarra (IdiSNA), Irunlarrea St., 3, 31008 Pamplona, Spain; 2Medical Oncology Department, Hospital Universitario de Navarra, Instituto de Investigación Sanitaria de Navarra (IdiSNA), 31008 Pamplona, Spain; 3Oncobiona Group, Navarrabiomed, Instituto de Investigación Sanitaria de Navarra (IdiSNA), 31008 Pamplona, Spain; 4Clinical Neuroproteomics Unit, Proteomics Platform, Proteored-ISCIII, Navarrabiomed, Instituto de Investigación Sanitaria de Navarra (IdiSNA), 31008 Pamplona, Spain; 5Gene Therapy and Regulation of Gene Expression, Centro de Investigación Médica Aplicada (CIMA), Instituto de Investigación Sanitaria de Navarra (IdiSNA), 31008 Pamplona, Spain; 6Medical Oncology Department, Hospital Clínico San Carlos, 28040 Madrid, Spain; 7Medical Oncology Department, Hospital Clínic i Provincial de Barcelona, IDIBAPS, 08036 Barcelona, Spain; 8Medical Oncology Department, Laboratory of Translational Genomics and Targeted Therapies in Solid Tumors, IDIBAPS, 08036 Barcelona, Spain

**Keywords:** lung cancer, NSCLC, immunotherapy, neutrophils, LDN, biomarkers, PD-1, immune checkpoint inhibitors

## Abstract

**Simple Summary:**

Immunotherapy has been positioned as frontline therapy for advanced non-small cell lung cancer (NSCLC), alone when PD-L1 tumor expression is high, or combined with chemotherapy otherwise. However, 50% of the patients do not respond to the treatment and the mechanisms of resistance are not well defined. Moreover, it is not clear whether chemo-immunotherapy could be advantageous in high PD-L1 tumor expression. We have found that baseline circulating low-density neutrophils (LDN) identify a subset of patients intrinsically refractory to immunotherapy. Interestingly, responses can be achieved with CT+IT, detecting a progressive depletion of LDN. Besides the potential role as predictive biomarker we observed that resistance was mediated by soluble molecules related with the HGF/c-MET pathway. Our findings establish circulating myeloid cells as one of the main mediators of resistance to immunotherapy in NSCLC, and give a rationale for potential drug combinations that might improve the outcomes.

**Abstract:**

Single-agent immunotherapy has been widely accepted as frontline treatment for advanced non-small cell lung cancer (NSCLC) with high tumor PD-L1 expression, but most patients do not respond and the mechanisms of resistance are not well known. Several works have highlighted the immunosuppressive activities of myeloid subpopulations, including low-density neutrophils (LDNs), although the context in which these cells play their role is not well defined. We prospectively monitored LDNs in peripheral blood from patients with NSCLC treated with anti-PD-1 immune checkpoint inhibitors (ICIs) as frontline therapy, in a cohort of patients treated with anti-PD1 immunotherapy combined with chemotherapy (CT+IT), and correlated values with outcomes. We explored the underlying mechanisms through ex vivo experiments. Elevated baseline LDNs predict primary resistance to ICI monotherapy in patients with NSCLC, and are not associated with response to CT+IT. Circulating LDNs mediate resistance in NSCLC receiving ICI as frontline therapy through humoral immunosuppression. A depletion of this population with CT+IT might overcome resistance, suggesting that patients with high PD-L1 tumor expression and high baseline LDNs might benefit from this combination. The activation of the HGF/c-MET pathway in patients with elevated LDNs revealed by quantitative proteomics supports potential drug combinations targeting this pathway.

## 1. Introduction

Immune checkpoint inhibitors (ICI) have improved the outcomes of patients diagnosed with advanced non-small cell lung cancer (NSCLC). Anti-PD1/PD-L1 antibodies have become the standard frontline treatment, as a single agent when PD-L1 tumor expression is high or combined with chemotherapy otherwise [1,2,3,4,5,6,7,8]. Unfortunately, the majority of the patients do not respond to treatment, no accurate predictive biomarkers able to identify those who will obtain a greater benefit have been discovered yet, and the mechanisms of resistance are not well understood [9,10]. Moreover, some patients with high PD-L1 expression who do not respond to single-agent ICI might benefit from chemoimmunotherapy combination.

Over recent years, the relevance of neutrophils in the immune response against cancer has been highlighted, although the understanding of their role is far from complete. Pro-tumor and anti-tumor activity have been described depending on the tissue and context [11,12]. A population of circulating neutrophils with immunosuppressive properties known as low-density neutrophils (LDNs) has been detected in cancer patients [13,14,15], although its association with response to ICI has not yet been defined. 

LDNs are an heterogeneous population comprising both mature and immature neutrophils. Whether it is a distinct lineage of neutrophils or the consequence of a premature release from bone marrow is still a matter of debate, and even the characteristics and function of LDNs might differ between cancer and other chronic inflammatory diseases, as even proinflammatory activities have been described in the latter [16]. However, a work by Sagiv JY et al. suggested a distinct origin and demonstrated that high density neutrophils (HDN) can switch to LDN through a TGF-β-dependent mechanism [13].

Our group previously demonstrated the association between circulating immune cells, particularly CD4 T cells, and the response to ICI in patients with NSCLC that have progressed to platinum-based chemotherapy [17]. Surprisingly, preliminary data from that project suggested that, in patients with advanced NSCLC receiving ICI as frontline treatment, circulating CD4 T cells did not have predictive value, while LDNs might be associated with primary resistance [18].

We have used flow cytometry to prospectively monitor LDNs levels in fresh blood samples from a cohort of patients with untreated advanced NSCLC receiving ICI monotherapy as frontline treatment, and evaluated the association with response and disease control. These findings were compared with a cohort of patients treated with chemoimmunotherapy combination. Lastly, we explored ex vivo the mechanisms of resistance using co-cultures, and compared the plasma of the different cohorts of patients using quantitative proteomics.

## 2. Materials and Methods

### 2.1. Study Design and Patient Enrolment

The study was approved by the Ethics Committee of the University Hospital of Navarre. Informed consent was obtained from all subjects and all experiments were performed according to the principles stablished in the Declaration of Helsinki and the Department of Health and Human Services Belmont Report. Samples were collected by the Blood and Tissue Bank of Navarre, Health Department of Navarre, Spain. Thirty-one patients diagnosed with NSCLC and PD-L1 tumor expression ≥50% were recruited at the University Hospital of Navarre. All received anti-PD1 immunotherapy (pembrolizumab) as frontline therapy according to current indications. Twenty patients diagnosed of NSCLC and PD-L1 0–49% were recruited at University Hospital of Navarre. They were treated with platinum-based chemotherapy combined with anti-PD1 immunotherapy (pembrolizumab) according to current indications. Exclusion criteria were mixed histologies, previous treatment for advanced disease or progression during neoadjuvant or adjuvant systemic treatment. Data from a cohort of healthy donors and of patients with NSCLC treated with anti-PD-1/PD-L1 after progression to first-line chemotherapy were also evaluated [17].

Ten ml of peripheral blood samples were obtained before the first cycle of immunotherapy, and after the first radiological control. PBMCs were isolated using Ficoll gradient as described elsewhere [17] and immune cell subpopulations were analyzed by flow cytometry. After centrifugation, LDN grouped at the interface between plasma and Ficoll as described [14], and were further analyzed along with PBMCs. Participation of each patient in the study concluded when a radiological test confirmed response or progression, or if the patient withdrew consent or died. Tumor responses were evaluated according to RECIST 1.1 [19] and Immune-Related Response Criteria [20]. Progressive disease was confirmed by at least one sequential tumor assessment, except in the case of clear clinical deterioration.

### 2.2. Flow Cytometry

Surface flow cytometry analyses were performed as described elsewhere [21]. Blood samples (10 mL) were obtained from each patient, and immediately processed. PBMCs were isolated by FICOL gradients. PBMCs were washed and cells were stained with the indicated antibodies in a final volume of 50 μL for 10 min in ice. The following fluorochrome-conjugated antibodies were used at 1:50 dilutions unless otherwise stated: CD3-APC (ref 130-113-135, Miltenyi Biotech, Bergisch-Gladbach, North Rhine-Westphalia, Germany), CD4-APC-Cy7 (ref 130-113-251, Miltenyi Biotech), CD4-FITC (ref 130-114-531, Miltenyi Biotech), CD8-FITC (ref 35-0088-T500, Tonbo Biosciences, San Diego, CA, USA), CD11b-PerCP-Cy5 (1:250) (ref 65-0112-U100, Tonbo Biosciences), CD14-PB (1:20) (ref 75-0149-T100, Tonbo Biosciences), CD27-PE (1:20) (ref 50-0279-T100, Tonbo Biosciences), CD28-PE-Cy7 (ref 302926, BioLegend, San Diego, CA, USA), CD28-PerPC-Cy5 (1:20) (ref 302921, BioLegend), CD45RA-PB (ref 130-113-922, Miltenyi Biotech), CD56-PE-Cy7 (ref 130-113-870, Miltenyi Biotech), CD57-PB (ref 130-123-866, Miltenyi Biotech), CD62L-APC (1:20) (ref 304810, BioLegend), CD62L-PerCP-Cy5 (ref 304823, BioLegend), CD66b-APC-Cy7 (ref 130-120-146, Miltenyi Biotech), CD95-FITC (ref 130-124-261, Miltenyi Biotech), CD116-APC (ref 130-100-986, Miltenyi Biotech), CD119-PE (ref 130-125-874, Miltenyi Biotech), KLRG1-APC-Cy7 (ref 130-120-563, Miltenyi Biotech), LAG3-PE (ref 369306, BioLegend) and PD1-PE-Cy7 (ref 130-120-391, Miltenyi Biotech). Gating strategy is presented in Appendix A.

### 2.3. Cell Culture

Human lung adenocarcinoma A549 cells were a kind gift of Prof Rubén Pío, authenticated by his group, and were grown in standard conditions. They were confirmed to be mycoplasma-free by PCR. These cells were modified with a lentivector encoding a single-chain version of a membrane-bound anti-OKT3 antibody [22]. The lentivector expressed the single-chain antibody construct under the control of the SFFV promoter and puromycin resistance from the human ubiquitin promoter in a pDUAL lentivector construct [23]. The single-chain antibody construct contained the variable light and heavy OKT3 immunoglobulin sequences separated by a G-S linker fused to a human IgG1 constant region sequence followed by the PD-L1 transmembrane domain.

Cell growth and cytotoxicity were monitored using xCELLigence real-time cell analysis (RTCA, Agilent Technologies, Santa Clara, CA, USA). Ten thousand A549-OKT3 cells resuspended in RPMI supplemented with 10% fetal bovine serum (FBS) were seeded in each well, adding plasma from patients or healthy donors at a 1:3 concentration when required by the experiment. After a 24-h incubation, 5000 T lymphocytes obtained from patients with NSCLC or healthy donors and stimulated with anti-CD3 and anti-CD28 were added. Each group had at least three repetitions, and each experiment was performed at least twice to confirm the results.

### 2.4. Data Collection and Statistics

Immune cell subpopulations were quantified using FlowJo (BD Biosciences, Franklin Lakes, NJ, USA). The percentage of LDNs were quantified prior to therapy (baseline) and after the first radiological control. Data were recorded by H.A., A.I.B., M.Z. and M.G., and separately analyzed by H.A. and M.G.

Treatments were administered to the patients according to current indications. Progression-free survival (PFS) was defined as the time from the starting date of therapy to the date of disease progression or death from any cause, whichever occurred first. PFS was censored on the date of the last patient consultation when no signs of progressive disease were evident. PFS was represented by Kaplan-Meier Plots and log-rank tests were used to compare cohorts. Derived neutrophil-to-lymphocyte ratio (dNLR) and Lung Immune Prognostic Index (LIPI) score were calculated as described [24]. Receiver operating characteristics (ROC) analysis were performed with baseline LDNs numbers, NLR, dNLR and neutrophils and disease control at 6 months yes/no as a binary output. Overall survival (OS) was defined as the time from the start date of therapy to the date of death from any cause. OS was evaluated in the same way as PFS.

Statistical tests were performed with GraphPad Prism 6 (GraphPad Software, San Diego, CA, USA) and SPSS statistical packages (IBM, Armonk, NY, USA). Percentages of LDNs were not normally distributed, so the comparisons between groups were made using Mann-Whitney and Kruskal-Wallis tests.

### 2.5. Proteomics

Plasma from the patients with NSCLC recruited for this study was purified before the administration of the first cycle of treatment, and frozen at −80 °C for further analysis. Plasma samples from patients with NSCLC that had progressed to platinum-based chemotherapy were recovered from a previous project [17]. The Multiple Affinity Removal Spin Cartridge System (Agilent Technologies, Miami, FL, USA) was used to remove the most abundant proteins according to the manufacturer’s instructions. Briefly, 6 μL of human plasma was diluted 16-fold with Buffer A and filtered through a 0.22-μm spin filter (1 min, 14,000× *g*). The non-bound protein fraction was collected and the column was washed twice with Buffer A and centrifuged (2.5 min, 100× *g*). Protein concentration was measured using Bradford assay kit (Bio-Rad, Hercules, CA, USA). MS/MS Library Generation and Quantitative Analysis were performed as previously described [25], and included 5 healthy donors, 4 untreated patients with NSCLC and high LDN levels, 5 untreated patients with non-squamous NSCLC and low LDN levels, 5 untreated patients with squamous NSCLC and low LDN levels and 5 patients with NSCLC that had progressed to platinum-based chemotherapy and high LDN levels. Briefly, 20 μg per sample were be used and the quantitative data obtained was analyzed using Perseus software (Max Planck Institute of Biochemistry, Munich, Germany) [26] for statistical analysis and data visualization.

### 2.6. Bioinformatics

Multiparametric flow cytometry data were represented in 2 dimensions using T-distributed Stochastic Neighbor Embedding (tSNE) algorithms [27]. Networks obtained from comparative proteomics of plasma from different cohorts were identified and represented using STRING (Search Tool for the Retrieval of Interacting Genes) software (http://stringdb.org/, accessed on several occasions from September 2021 to March 2022) [28].

## 3. Results

### 3.1. Study Population

Thirty-one patients diagnosed with advanced NSCLC who received pembrolizumab as frontline therapy were recruited. All had PD-L1 tumor expression ≥ 50%. Baseline characteristics are depicted in Table 1. Overall response rate (ORR) was 42.9%, and disease control rate (DCR) was 64.3%. Median PFS and OS were 5.7 and 33.0 months, respectively. Fast progressive disease (fast-PD)/early death rate, defined as death at 12 weeks after starting, was 25.8%.

In another study cohort, 21 patients diagnosed with advanced NSCLC who received chemotherapy-immunotherapy combination (CT+IT) including pembrolizumab as first-line treatment were recruited. Baseline characteristics are depicted in Table 2. Overall, 61.9% had PD-L1 expression ≤ 1%, 14.3% had PD-L1 expression 1–4% and 23.8% had PD-L1 expression 5–49%. ORR was 47.4%, and DCR was 52.6%. Median PFS was 3.6 months, while median OS was not reached. The rate of fast-PD/early death was 9.5%.

### 3.2. Baseline Low-Density Neutrophils (LDNs) and Response to ICI Monotherapy as Frontline Treatment in NSCLC

In the cohort of 31 patients diagnosed with advanced NSCLC who received pembrolizumab as frontline therapy, the overall immune cell composition in peripheral blood was characterized by high-dimensional flow cytometry and tSNE analyses in 3 consecutive patients with disease control longer than 6 months. These results were compared to those of 3 consecutive progressors. To avoid the confounding effect of chemotherapy, initially only patients receiving pembrolizumab monotherapy were evaluated.

Previous work by our group showed that relative percentages of highly differentiated CD4 T cells (CD4 THD) in peripheral blood were a good predictive biomarker of response to second-line ICI monotherapies [17]. However, no correlation was observed in patients treated with frontline ICI therapy. Interestingly, a strong enrichment of low-density neutrophils (LDNs), a subpopulation with immunosuppressive properties identified as CD11b+ CD116+ CD66b+ CD3− CD14−, was found in progressors, while this population was apparently absent in responders (Figure 1A).

In the overall cohort of patients treated with ICI monotherapy, higher levels of LDNs were detected in progressors compared to responders or healthy donors (mean 25.2%, 2.7% and 0.7%, respectively; *p* < 0.0001). No differences were found according to age or tumor burden (Appendix A). To evaluate the value of baseline LDNs as a biomarker of response, ROC analyses were performed and an area under the curve (AUC) of 0.908 (*p* < 0.001) was calculated. ROC analyses established a LDN threshold of 7.09%, which identified patients who showed disease control of less than 6 months with a sensitivity of 84.6% and specificity of 93.3% (Figure 1B). Patients with LDNs above this threshold presented an ORR of 0%, lower median progression free survival (mPFS) (5.6 weeks vs. non-reached, *p* < 0.001) and lower median OS (mOS) (6.1 weeks vs. non reached, *p* = 0.004) (Figure 1C). Furthermore, the incidence of fast-PD/early death was significantly higher in patients with NSCLC and high baseline LDNs (66.7% vs. 6.3%, *p* = 0.001) (data not shown).

### 3.3. High LDN Levels and Resistance to First-Line Chemoimmunotherapy in Patients with NSCLC

Twenty-one patients receiving chemoimmunotherapy combination (CT+IT) containing pembrolizumab as first-line treatment were recruited. In contrast to the results from first-line ICI monotherapy, baseline proportions of circulating LDNs did not predict response to CT+IT (AUC 0.350, *p* = 0.257), and levels did not differ between responders and non-responders (mean 31.2% and 16.8%, *p* = 0.14), suggesting a predictive and not prognostic value of LDNs (Figure 2A). Moreover, some patients with high baseline LDN levels (*n* = 9) responded to treatment and we observed in all of them a brisk decline in LDNs between the first and the second cycle. Significant proportions of LDNs were no longer observed at the time of the first radiological follow-up. This suggests that these cells play an active role in ICI resistance (Figure 2B).

To confirm that the predictive value of LDNs was specific to patients treated with ICI monotherapy as first-line therapy, we retrospectively studied the flow cytometry staining from a previous well-characterized cohort of patients with NSCLC treated with ICI monotherapy after progression to platinum-based chemotherapy [17]. No significant differences in LDNs were found between first-line and pretreated NSCLC (median 8.1% vs. 4.4%, *p* = 0.15) (data not shown). Again, the mean proportion of LDNs was not higher in patients who progressed to treatment compared with responders (9.5% vs. 19.1%, *p* = 0.13) (data not shown).

Previous works have demonstrated the predictive value of baseline neutrophils in peripheral blood tests from patients with NSCLC treated with ICI [24,29,30]. To identify any relationship between neutrophils and LDNs, we studied the correlation between neutrophils from ordinary blood samples and LDNs quantified by flow cytometry. A statistically significant association was found by linear regression (*p* = 0.047), although the strength of the association was low. We observed that all patients with high levels of neutrophils presented a high proportion of LDNs (Figure 2C). However, 54% of patients with LDNs above the 7.08% threshold had no neutrophilia, suggesting that the expansion of LDN is independent of peripheral neutrophils.

The predictive value of peripheral neutrophils, neutrophil-to-lymphocyte ratio (NLR) and derived NLR (dNLR) was then studied. These variables were associated with progression to ICI monotherapy, although with lower sensitivity and specificity than LDNs (AUC 0.638, *p* = 0.11; AUC 0.750, *p* = 0.02; AUC 0.634, *p* = 0.21) (Figure 2D). Interestingly, NLR and dNLR were also associated with resistance to CT+IT (Figure 2E), and with OS in both the ICI monotherapy and the CT+IT cohort (not shown), indicating prognostic and not predictive value.

### 3.4. Soluble Factors in Plasma from Patients with High LDN Levels Impair Antitumor Immune Response Ex Vivo

To evaluate ex vivo T cell effector functions from the different cohorts of patients, we performed co-cultures of T cells with A549-SC3 cells, a human lung cancer cell line engineered by us to express membrane-bound anti-CD3 single-chain antibody to ensure tumor cell recognition by lymphocytes independently of antigen-specificity [17]. T cells from healthy donors were used as controls. T cells from patients with NSCLC regardless of their LDN status showed comparable cytotoxic activities to T cells from healthy donors (Figure 3A). These results showed that T cells from these patients are not dysfunctional per se.

It had been previously demonstrated that soluble factors might mediate immune suppression caused by LDNs [31]. To find out if this was the case, co-cultures of A549-SC3 cells with T cells isolated from healthy donors were carried out in the presence or absence of plasma from healthy donors, from patients with NSCLC and normal LDNs proportions, and from patients with NSCLC and elevated LDNs. Interestingly, only plasma from patients with elevated LDNs levels abrogated T cell cytotoxicity over A549-SC3 cells (Figure 3B). Furthermore, in the absence of T cells, the plasma of these patients promoted cancer cell growth (Figure 3C).

Finally, to identify potential immunosuppressive candidates in plasma from patients with NSCLC and elevated LDNs, we performed quantitative proteomics in plasma. To this end, plasma from healthy donors, from untreated NSCLC patients with normal LDN proportions, from untreated patients with NSCLC and elevated LDNs, and from patients with NSCLC and elevated LDNs treated with ICI monotherapy after progression to platinum-based chemotherapy were used.

Interestingly, untreated patients with NSCLC and elevated LDNs exhibited a distinct proteome. We found an enrichment in proteins related with neutrophil polarization and regulation of inflammation, suggesting a potential role for LDNs in therapeutic resistance. Among these, hepatocyte-growth factor (HGF) activator (HGFAC), a serin-protease that activates HGF, was significantly elevated (Figure 3D). HGF is an activator of the MET pathway, which has been associated with peripheral expansion of neutrophils [32], as well as with immunosuppressive activity of these cells and lack of tumor T lymphocyte infiltration [31]. We quantified plasmatic HGF by ELISA and found a trend towards higher levels in patients with NSCLC compared with controls (median 550.2 and 346.3 pg/mL, *p* = 0.08). However, no differences were observed between patients with or without elevated LDNs (mean 915.4 and 799.4 pg/mL, *p* = 0.7) (Appendix A).

## 4. Discussion

The precise mechanisms underlying the anticancer immune response unleashed by immunotherapy are still not well characterized. Even though several resistance mechanisms have been described, their impact on the efficacy of the treatment in the different cancer subtypes is yet to be defined. Several papers exploring predictive biomarkers of response to ICI in NSCLC have been published up to date. However, the actual role that these biomarkers might play in the immune response has not been sufficiently detailed, probably due to the recruitment of heterogeneous cohorts of patients receiving different treatments in dissimilar clinical settings.

In agreement with other studies [33,34,35], we found a strong association between high baseline levels of circulating LDNs and resistance to ICI in untreated patients. LDN levels are higher in patients with advanced cancer compared with early-stage cancer patients or healthy controls [33,34]. Interestingly, LDNs were not associated with resistance to CT+IT, which indicates not only that circulating LDNs are a predictive biomarker of resistance to single-agent ICI as frontline therapy, but also suggests that chemotherapy combined with immunotherapy might deplete this cell population and allow an antitumor response. Neutrophil depletion, as well as IL-6 blockade, were associated with enhanced anti-PD1 immunotherapy efficacy in 2 lung cancer murine models [36,37]. Accordingly, we found that all patients with elevated LDNs who responded to CT+IT showed a brisk decline of LDNs after the first cycle, and this population was almost absent at the first radiological follow-up.

High baseline levels of neutrophils in routine blood tests, often represented as NLR or dNLR, has been stablished as an adverse biomarker, both prognostic and predictive of response to ICI [24]. We observed a low strength association between neutrophils in blood tests and LDN measured by flow cytometry, suggesting that some patients present a generalized expansion of myeloid subsets while some tumors can induce a specific expansion of LDNs. A recent paper reported higher rates of fast-PD/early death in patients with NSCLC and high dNLR before starting ICI monotherapy [38]. We found a numerically higher proportion of fast-PD/early death in the ICI monotherapy cohort than in the CT+IT cohort, and an association between fast-PD/early death and high LDNs in the ICI monotherapy cohort but not in the CT+IT cohort. This suggests that CT+IT treatment could prevent fast-PD/early death in patients with NSCLC and high baseline LDNs receiving ICI monotherapy.

A translational study found that high blood neutrophil counts in melanoma patients refractory to immunotherapy were associated with high serum HGF levels. The authors demonstrated in a murine model that the inhibition of the HGF/c-MET axis impaired the recruitment of immunosuppressive neutrophils into tumors, thus allowing T cell tumor infiltration and enhancing the effect of immunotherapy [31]. Using ex vivo co-cultures we observed that plasma from patients with NSCLC and elevated LDN levels impaired the cytotoxic activities of T cells and promoted tumor cell proliferation. We used quantitative proteomics to compare the plasma of patients with elevated LDNs, and found in this subset of patients a specific upregulation of proteins associated with the immunosuppressive role of LDNs, the HGF/c-MET pathway being a potential route involved. Proteomics studies from tissue samples would have probably provided relevant additional data, but unfortunately in most of the cases these samples were not available as they were used for routine diagnostic test required for clinical practice. We did not find higher HGF levels in plasma from patients with high LDNs, which might reflect that the activation of the HGF/c-MET pathway depends on the overexpression of HGFAC rather than on increased production of HGF.

Our results show a specific association between LDNs and resistance to ICI monotherapy as frontline treatment in NSCLC. Patients with high baseline LDN levels, regardless of high PD-L1 tumor expression, might benefit from CT+IT. Even though LDNs appear to have a major role in the resistance to ICI monotherapy in NSCLC, other factors such as tumor gene alterations [39,40], additional infiltrating [41] or circulating immune cells [42], tumor microenvironment elements [43], as well as host related factors including intestinal microbiota [44], could be influencing the results. The spontaneous plasticity of LDNs might as well be a confounding factor, as the levels of these cells could significantly change over time and impact the efficacy of ICI [13]. Moreover, some of the previously mentioned factors also have a role in neutrophil differentiation and plasticity, and might indirectly affect the outcomes through mechanisms yet unexplored [45,46,47]. Due to the exploratory nature of the study, these findings will be validated in an independent cohort that is currently under recruitment. The potential benefit of ICI combining with HGF/c-MET pathway inhibitors could be explored in further studies.

## 5. Conclusions

Baseline LDNs identify a subset of patients with NSCLC intrinsically refractory to ICI monotherapy as frontline treatment. The combination of chemotherapy with ICI causes a depletion of LDNs, thus eliciting the antitumor immune response, and preventing fast-PD/early death. LDNs mediate resistance to ICI through humoral mechanisms, mainly via the HGF/c-MET pathway, suggesting a potential synergy between ICI and c-MET targeting drugs.

## Figures and Tables

**Figure 1 cancers-14-03846-f001:**
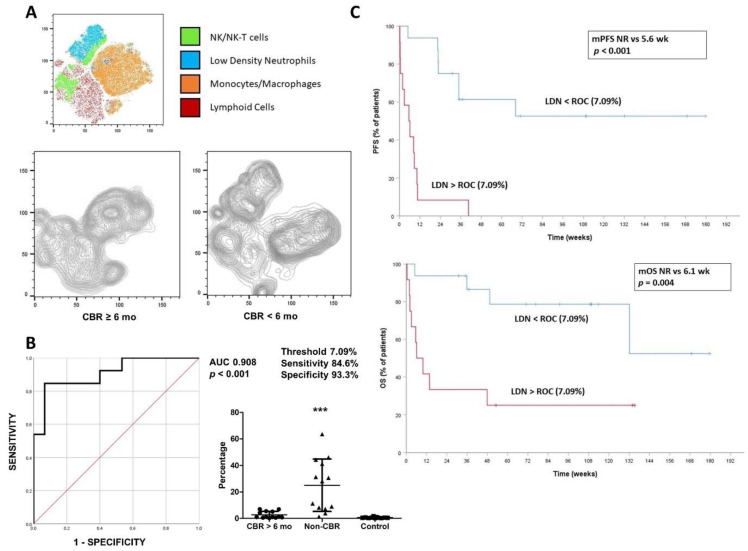
(**A**): Top, tSNE graph of myeloid subpopulations represented by different colors. NK/NK-T cells: CD11b+, CD56+, CD14−, CD66b−. LDNs: CD11b+, CD66b+, CD116+, CD14−. Monocytes/macrophages: CD11b+, CD14+, CD116+, CD66b−. Lower left, tSNE graph of 3 responders to immune checkpoint inhibitors (ICI) monotherapy. Lower right, tSNE graph representative of 3 non-responders to ICI monotherapy. (**B**): Left, ROC analysis of baseline low-density neutrophils (LDNs) as a function of clinical benefit rate (CBR) < 6 months. Right, baseline levels of LDN in patients receiving ICI monotherapy with CBR longer than 6 months compared with patients with CBR less than 6 months and with healthy donors. (**C**): Top, progression free survival stratified by the presence of baseline LDNs above the ROC threshold (7.09%). Below, overall survival stratified by the presence of baseline LDN above the ROC threshold. *** indicate highly significant (*p* < 0.001) statistical differences.

**Figure 2 cancers-14-03846-f002:**
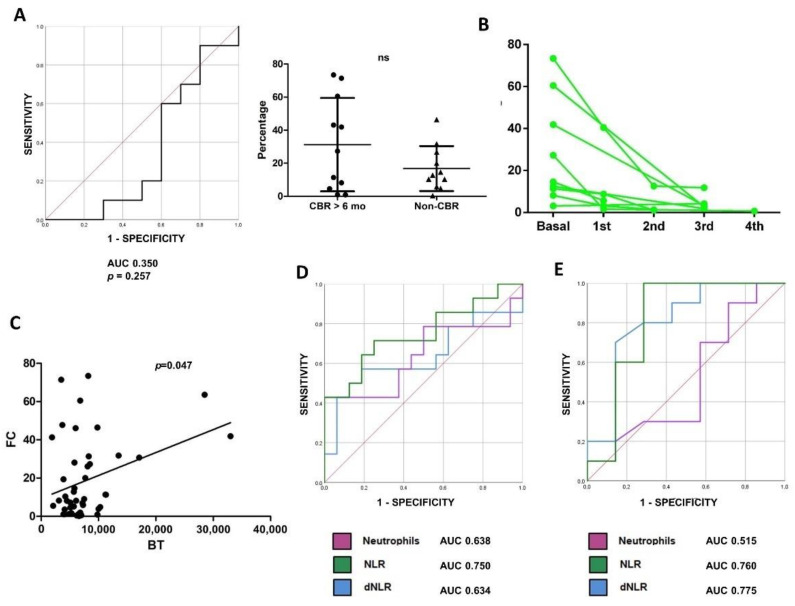
(**A**): Left, ROC analysis of baseline LDNs as a function of CBR < 6 months in patients receiving chemoimmunotherapy (CT+IT). Right, baseline levels of LDN in patients receiving CT+IT with CBR for more than 6 months compared to patients with CBR less than 6 months. (**B**): Monitoring of patients with LDNs above the threshold who responded to CT+IT. (**C**): Scatter plot representing the association between baseline LDNs and baseline neutrophils in blood test in the whole group of patients. (**D**): ROC analysis of baseline neutrophils, neutrophil-to-lymphocyte ratio (NLR) and derived neutrophil-to-lymphocyte ratio (dNLR) as a function of CBR < 6 months in patients receiving ICI monotherapy. (**E**): ROC analysis of baseline neutrophils, NLR and dNLR as a function of CBR < 6 months in patients receiving CT+IT. ns indicate non-significant statistical differences.

**Figure 3 cancers-14-03846-f003:**
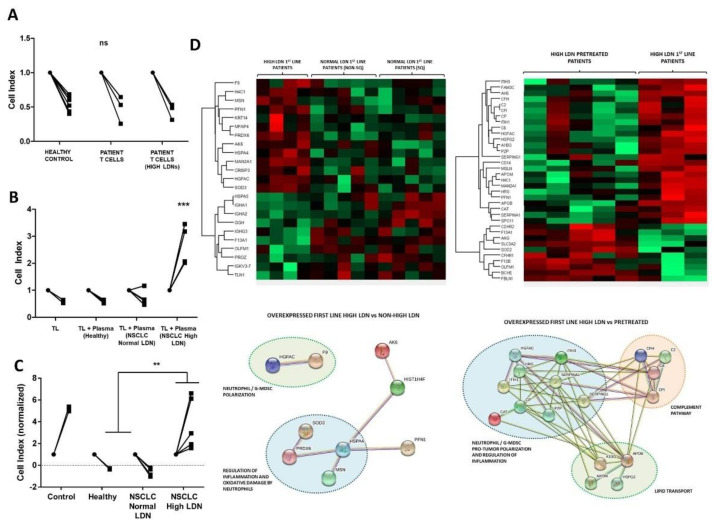
(**A**): Cell index of A549-OKT3 cells before and after the addition of T lymphocytes from the cohorts indicated below. (**B**): Cell index before and after the addition of T lymphocytes from a healthy donor and plasma from the cohorts indicated below. (**C**): Cell index before and after the addition of plasma from the cohorts indicated below. (**D**): Upper left, heat-map comparing the complete identified proteome from the plasma of untreated NSCLC patients with high baseline LDN levels, of untreated squamous NSCLC patients with normal LDN levels and of untreated non-squamous NSCLC patients with normal LDN levels. Top right, heatmap comparing the complete identified proteome from the plasma of untreated NSCLC patients with high baseline LDN levels, with that of NSCLC patients with high LDN levels who progressed to platinum-based chemotherapy. Lower left, functional interactomes with significantly upregulated proteins in untreated NSCLC patients with high baseline LDN levels compared to untreated NSCLC patients with normal baseline LDN levels. Lower right, functional interactomes with significantly upregulated proteins in untreated NSCLC patients with high baseline LDN levels compared to NSCLC patients with normal baseline LDN levels who progressed on platinum-based chemotherapy. **; ***, indicate very significant (*p* < 0.01) and highly significant (*p* < 0.001) statistical differences respectively. Ns indicate non-significant statistical differences.

**Table 1 cancers-14-03846-t001:** Baseline characteristics of the ICI monotherapy cohort.

Variable		Percentage
Age	<70 ≥70	21 (67.7%)
		10 (33.3%)
Sex	Female Male	7 (22.6%)
		24 (77.4%)
Performance Status	0–1 2–4	27 (87.1%)
		4 (12.9%)
Histology	Non-squamous Squamous	22 (71%)
		9 (29%)
Stage	Stage IIIA-C Stage IV	3 (9.7%)
		28 (90.3%)
Tumor burden	Less than 3 organs 3 organs or more	12 (38.7%)
		19 (61.3%)
Liver metastases	Yes No	8 (25.8%) 23 (74.2%)
PD-L1 tumor expression	≥50%	31 (100%)
Neutrophil-to-lymphocyte ratio (NLR)	≤6 >6 Unknown	21 (67.7%) 7 (22.6%) 3 (9.7%)
Serum lactate dehydrogenase (LDH)	≤upper limit of normal >upper limit of normal Unknown	10 (32.3%) 8 (25.8%) 13 (41.9%)
Serum albumin	≥3.5 g/dL <3.5 g/dL Unknown	20 (64.5%) 8 (25.8%) 3 (9.7%)
Gustave Roussy Immune Score (GRIm)	0–1 2–3 Unknown	17 (54.8%) 6 (19.4%) 8 (25.8%)
Derived neutrophil-to-lymphocyte ratio (dNLR)	≤3 >3 Unknown	25 (80.6%) 5 (16.1%) 1 (3.2%)
Lung Immune Prognostic Index (LIPI)	Good/Intermediate Poor Unknown	28 (90.3%) 2 (6.5%) 1 (3.2%)

**Table 2 cancers-14-03846-t002:** Baseline characteristics of the CT+IT cohort.

Variable		Percentage
Age	<70 ≥70	17 (85.0%)
		3 (15.0%)
Sex	Female Male	5 (25.0%)
		15 (75.0%)
Performance Status	0–1 2–4	18 (90.0%)
		2 (10.0%)
Histology	Non-squamous Squamous	18 (90.0%)
		2 (10.0%)
Stage	Stage IIIA-C Stage IV	1 (5.0%)
		19 (95.0%)
Tumor burden	Less than 3 organs 3 organs or more	5 (25.0%)
		15 (75.0%)
Liver metastases	Yes No	6 (30.0%) 14 (70.0%)
PD-L1 tumor expression	0% 1–4% 5–49%	13 (65.0%) 2 (10.0%) 5 (25.0%)
Neutrophil-to-lymphocyte ratio (NLR)	≤6 >6	11 (55.0%) 9 (45.0%)
Serum lactate dehydrogenase (LDH)	≤upper limit of normal >upper limit of normal Unknown	5 (25.0%) 7 (35.0%) 8 (40.0%)
Serum albumin	≥3.5 g/dL <3.5 g/dL Unknown	18 (90.0%) 1 (5.0%) 1 (5.0%)
Gustave Roussy Immune Score (GRIm)	0–1 2–3 Unknown	13 (65.0%) 5 (25.0%) 2 (10.0%)
Derived neutrophil-to-lymphocyte ratio (dNLR)	≤3 >3	12 (60.0%) 8 (40.0%)
Lung Immune Prognostic Index (LIPI)	Good/Intermediate Poor Unknown	15 (75.0%) 3 (15.0%) 2 (10.0%)

## Data Availability

Not applicable.

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
