# Peer review of "Circulating Low Density Neutrophils Are Associated with Resistance to First Line Anti-PD1/PDL1 Immunotherapy in Non-Small Cell Lung Cancer"

_cancers, 2022, doi:10.3390/cancers14163846_

Round 1

Reviewer 1 Report

In this manuscript, authors report circulating low-density neutrophils (LDN) is associated to NSCLC patients’ resistance to immunotherapy and found enrichment of proteins HGF/c-MET pathway contributes to the immunosuppressive role of LDNs. Overall it’s a good paper.

Here are a few concerns:

1. Some previous papers have already reported the association between high circulating LDNs and resistance to ICI in patients. What is the novelty of your manuscript?

2. All NSCLC tumor patients have similar level of PDL1 expression?

3. Besides LDN, what other factors could contribute to resistance to the anti-PD1/PDL1 immunotherapy in NSCLC? Could you also discuss about it in the discussion section?

4. Figure 1A images are hard to recognize, could you please enlarge it and also the le

5. Typo: line 259, the expansion LDN expansion.

Reviewer 2 Report

Manuscript ID: Cancers-1810969

Title: “Circulating Low-Density Neutrophils Are Associated with Resistance to First Line Anti-PD1/PDL1 Immunotherapy in Non-Small Cell Lung Cancer’’

In the manuscript, Arasanz et al. found that baseline levels of circulating low-density neutrophils (LDN) stratify patients with immunotherapy resistance. They discovered the HGF/c-MET pathway as a factor in immunotherapy resistance. The study's topic is important and valuable because, despite major advancements in lung cancer treatment, long-term survival is still rare, and a deeper understanding of molecular phenotypes would allow the identification of new therapeutic opportunities and mechanisms of resistance. Overall, the paper is timely and well-presented. However, I have some conceptual and technical concerns that should be addressed to create a stronger paper that more clearly demonstrates its “innovation” in the field prior to publication.

Comments:

1.      No rationale is provided for investigating low-density neutrophils (LDN) in first place. Indeed, the author discovered strong enrichment of LDN in progressors, but this does not explicitly justify their rationale. What about the presence of macrophages, which have been linked to immunotherapy resistance in NSCLC and are present in Figure 1A?

2.  What clinicopathological inclusion/exclusion criteria were used to stratify the clinical samples. Please provide the information.

3.      No information is provided on how sample size calculation is performed making it difficult to assess statistical validity.

4.      Did a subset of these identified proteins found to be detectable and differentially present in the peripheral blood of cases and controls with or without anti-PD1 immunotherapy? Please provide information.

5.      To verify the accuracy of the proteomics analysis, it is imperative to perform an immunohistochemical analysis of key proteins in clinical samples to validate their findings.

6.      The AUC for baseline neutrophils, neutrophil-to-lymphocyte ratio (NLR) and derived neutrophil-to-lymphocyte ratio (dNLR) in NSCLC patients receiving anti-PD1 monotherapy does not have high diagnostic accuracy. What would the author's explanation be? Please provide rational justifications.

7. The authors have provided a minimal amount of information about the multiparametric flow cytometry data using tSNE. To rule out any discrepancy, the author is advised to show the gating strategy used to profile overall immune cell composition in peripheral blood of patients who received pembrolizumab as frontline therapy as supplemental data.

8.  It will be interesting to investigate how combining a clinical marker of host immunity (NLR, LDN) with a genomic marker of tumor antigenicity predicts survival and response rates.

9.    Do LDN represent different stages of differentiation of the neutrophils or are they distinct neutrophil subpopulations. What about their cellular origin?

10.  What mediates the change in neutrophil density? Is it a tumor or immunotherapy treatment?

11.  What is the level of PDL1 on LDN compared to tumor cells? Please show data.

12.  Is LDN found in healthy donors?

13.  Is LDN derived from patient blood cytotoxic to tumors and T cells? Please show evidence to correlate the findings.

14.  How do the tumor stage and age relate to LDN levels?

15.  The flow-tSNE data lacks appropriate clarity and transparency. Based on Figure 1A, it is not clear whether the difference between the responder and non-responder is due to batch effects or not. The enriched genes are not legible in the figure.

16.  The authors need to show gene expression of canonical neutrophil, macrophage, NK cells, etc associated genes across the clusters to verify that they have indeed successfully profiled neutrophils in both responder and non-responder NSCLC patient blood

17.  On page 10, line 311, the author stated that HGF is an activator of the MET pathway, which has been linked to peripheral neutrophil expansion and cited the paper. However, according to the cited study, MET is required for the recruitment of anti-tumoral neutrophils. If this is the case, how can the author claim that LDN is immunosuppressive? The authors need to provide additional data or revise their claims throughout the manuscript.

18.  The source and data for healthy controls are missing to assess the findings clinically.

19.  Author state that … We found enrichment in proteins related with neutrophil polarization and regulation of inflammation in untreated patients with NSCLC and elevated LDNs. However, no clinical evidence exists to support the role of HGF signaling in LDN. Furthermore, the source and number of untreated NSCLC patients are unknown.

20.  The author used the terms high baseline LDN and high LDN multiple times. It's somewhat deceptive and perplexing. Please make logical corrections.

21.  How does the author define normal and high LDN?

22.  Results are poorly discussed and clinically evaluated. The author needs to re-evaluate their findings by clinically correlating with existing scientific literature and presented in a more refined scientific manner.

23.  The discussion is not explanatory enough, lacks an in-depth mechanistic approach, and most importantly, didn’t critically discuss existing findings.

24.  The supporting data is also not available to review, making it impossible to identify how robust it is. 

Reviewer 3 Report

By analyzing fresh blood samples from a cohort of patients with advanced NSCLC treated with ICI monotherapy and evaluating the association with response and disease control, the author found that baseline circulating low-density neutrophils (LDN) identify a subset of patients intrinsically refractory to immunotherapy. CT+IT which depleted LDN was able to overcome the resistance. By performing proteomics, the authors found that the HGF/c-MET pathway in patients with elevated LDNs was activated, which has been associated with peripheral neutrophil expansion, and immunosuppressive activity. The experiments were designed and conducted properly. The manuscript was well written well and easy to follow.

One minor concern is the characteristic of LDN is controversial, some studies think they are more like mature neutrophils, in this paper, proteomics indicated that LDN seems to share features like MDSC, which has a strong immunosuppressive function. It is better to co-culture LDN with T cells in vitro to see if LDN is able to suppress T cell activation and functions.

Round 2

Reviewer 2 Report

Comments:

1.      How did the author separate the LDN fraction from the PBMC, which contained HDN (High density) and NDN (normal density) fractions? Did the author use centrifugation before performing surface flow cytometry analyses? Please provide evidence and describe the procedure of LDN isolation in the method section with proper citation.

2.      How does the author reconcile their findings with a report (PMID: 29207617) showing a significant LDN fraction in healthy controls in lung cancer settings?

3.      How does the degree of spontaneous plasticity in LDN and HDN affect therapy outcomes in lung cancer? Please discuss.

4.      Please correct ref. 15 on page 2; line 17 to match the statement.

Author Response

Again, thank you very much for the detailed review and constructive comments.
